# RecA and Specialized Error-Prone DNA Polymerases Are Not Required for Mutagenesis and Antibiotic Resistance Induced by Fluoroquinolones in *Pseudomonas aeruginosa*

**DOI:** 10.3390/antibiotics11030325

**Published:** 2022-02-28

**Authors:** Jessica Mercolino, Alessandra Lo Sciuto, Maria Concetta Spinnato, Giordano Rampioni, Francesco Imperi

**Affiliations:** 1Department of Science, Roma Tre University, 00146 Rome, Italy; jes.mercolino@stud.uniroma3.it (J.M.); alessandra.losciuto@uniroma3.it (A.L.S.); mariaconcetta.spinnato@uniroma3.it (M.C.S.); giordano.rampioni@uniroma3.it (G.R.); 2IRCCS Fondazione Santa Lucia, 00179 Rome, Italy

**Keywords:** DinB, homologous recombination, ImuBC, induced mutagenesis, PolB, SOS response, resistance

## Abstract

To cope with stressful conditions, including antibiotic exposure, bacteria activate the SOS response, a pathway that induces error-prone DNA repair and mutagenesis mechanisms. In most bacteria, the SOS response relies on the transcriptional repressor LexA and the co-protease RecA, the latter being also involved in homologous recombination. The role of the SOS response in stress- and antibiotic-induced mutagenesis has been characterized in detail in the model organism *Escherichia coli*. However, its effect on antibiotic resistance in the human pathogen *Pseudomonas aeruginosa* is less clear. Here, we analyzed a *recA* deletion mutant and confirmed, by conjugation and gene expression assays, that RecA is required for homologous recombination and SOS response induction in *P. aeruginosa*. MIC assays demonstrated that RecA affects *P. aeruginosa* resistance only towards fluoroquinolones and genotoxic agents. The comparison of antibiotic-resistant mutant frequency between treated and untreated cultures revealed that, among the antibiotics tested, only fluoroquinolones induced mutagenesis in *P. aeruginosa*. Notably, both RecA and error-prone DNA polymerases were found to be dispensable for this process. These data demonstrate that the SOS response is not required for antibiotic-induced mutagenesis in *P. aeruginosa*, suggesting that RecA inhibition is not a suitable strategy to target antibiotic-induced emergence of resistance in this pathogen.

## 1. Introduction

The concurrence of genetic variation and natural selection is a fundamental force driving the evolution of all organisms, including bacteria. While the DNA replication process unavoidably causes spontaneous mutations during bacterial growth, the so-called “adaptive mutations” are induced by external stimuli and help bacteria to cope with stressful conditions [1,2]. One of these conditions is antibiotic exposure, which can increase the DNA mutation rate either by selecting cells with loss-of-function mutations in repair systems (hypermutator variants), or by directly promoting the expression and/or activity of error-prone DNA repair systems [3]. Antibiotic-induced mutagenesis is thought to be clinically important, as it can promote the emergence of resistant mutants during antibiotic treatment, thus contributing to the spread of drug resistance in bacterial pathogens [4,5].

In many bacteria, stress- and antibiotic-induced mutagenesis involves the SOS response. This is a finely regulated pathway that controls the expression of gene products that delay cell division and promote mutations, recombination and/or DNA repair in response to DNA damage [6,7]. In the model bacterium *Escherichia coli*, the activation of the SOS response leads to the induction of over 50 genes, referred to as SOS genes. During normal growth, the transcriptional repressor LexA represses SOS genes by binding to a specific 20-bp operator sequence (the SOS box) in their promoters [8]. DNA damage and replication fork stalling cause the accumulation of single-stranded DNA (ssDNA), which in turn leads to the activation of the recombination protein RecA [8,9]. RecA forms RecA-ssDNA nucleoprotein filaments that, by inducing LexA self-cleavage, lead to the derepression of SOS genes, including error-prone DNA repair systems and error-prone DNA polymerases [9,10]. The action of these polymerases, also known as translesion synthesis (TLS) polymerases, renders the DNA replication and repair processes more erroneous, thus increasing the mutagenesis rate [3,11,12].

In several bacterial species, stress caused by antibiotic exposure increases the emergence of antibiotic resistant clones by inducing the SOS response [4,5,13,14]. Given the central role of RecA in the activation of the SOS response, this protein has been exploited as a potential target for the development of drugs able to reduce the evolution of antibiotic resistance [15,16].

Although some SOS inhibitors have been found to prevent the self-cleavage of the LexA ortholog of the human pathogen *Pseudomonas aeruginosa in vitro* [17], the SOS response and its role in adaptive mutagenesis in this bacterium have been poorly investigated so far. *P. aeruginosa* is a major opportunistic pathogen responsible for a broad range of infections in humans, especially in immunocompromised and cystic fibrosis (CF) patients [18,19]. Its high antibiotic resistance, mostly due to low outer membrane permeability and active drug efflux, makes *P. aeruginosa* infections challenging to treat [20]. The evolution of resistance in this pathogen is particularly problematic in CF patients, whose recurrent lung infections require prolonged therapeutic treatments. These ultimately select for CF lung-adapted mutants that establish chronic infections, leading to lung failure [21,22]. RecA and LexA orthologs are also major players of the SOS response in *P. aeruginosa*. Transcriptomic analysis revealed that the *P. aeruginosa* and *E. coli* LexA regulons are functionally similar, comprising error-prone polymerase and DNA repair genes [23]. That said, differences in the SOS response in *P. aeruginosa* and *E. coli* exist. As an example, beside LexA, *P. aeruginosa* RecA promotes the self-cleavage of two additional transcriptional regulators, i.e., PrtR and PA0906. PA0906 controls the expression of a few genes of unknown function, while PrtR represses the expression of genes required for the biosynthesis and secretion of pyocins, antimicrobial polypeptides used for intra- and inter-species competition [24].

A number of studies have confirmed that, as for *E. coli*, exposure to the fluoroquinolone antibiotic ciprofloxacin induces the SOS response in *P. aeruginosa* [23,25,26,27], and increases its mutation rate [28,29,30,31]. However, the causal link between RecA or the SOS response and acquired antibiotic resistance is still unclear in this bacterium. Moreover, it has been reported that RecA can also increase the mutation rate in an SOS-independent manner through the mutagenic recombinational repair of double strand breaks (DSBs), although this mechanism was mainly observed in *P. aeruginosa* biofilms rather than in planktonic cells [29,32]. Finally, the impact of RecA on the intrinsic level of resistance to antibiotics other than fluoroquinolones has not been investigated in *P. aeruginosa* yet. Thus, this work aimed at exploring the effect of RecA depletion on both intrinsic and acquired antibiotic resistance in *P. aeruginosa*. This information is important to predict the suitability of RecA inhibition for the development of novel antipseudomonal drugs and/or antibiotic adjuvants reducing the spread of antibiotic resistance in *P. aeruginosa*.

## 2. Results and Discussion

### 2.1. Validation of the P. aeruginosa RecA-Deficient Mutant as an SOS Response- and Homologous Recombination-Null Strain

The role of RecA in homologous recombination and recombinational DNA repair in the model organism *E. coli* has been widely explored [33,34]. Accordingly, it has been demonstrated that the lack of RecA in *E. coli* leads to recombination deficient strains [35]. To confirm that RecA is essential for homologous recombination also in *P. aeruginosa*, the acquisition efficiency of three different mobilizable plasmids, pFLP2, mini-CTX1, and pDM4Δ*rsmA* was compared between the wild type strain PAO1 and an isogenic Δ*recA* mutant. Plasmids used in the conjugation experiments were chosen based on the different strategies they use to be stably maintained in *P. aeruginosa* cells. While pFLP2 is a self-replicating vector in *P. aeruginosa* [36], the other two plasmids are suicide vectors, able to replicate in *E. coli* but not in *P. aeruginosa* unless they integrate into the chromosome through site-specific recombination (mini-CTX1), mediated by a plasmid-encoded recombinase [37], or through homologous recombination (pDM4Δ*rsmA*) between DNA regions present both in the plasmid and in the *P. aeruginosa* chromosome [38]. As shown in Figure 1a, the frequency of transconjugants, corresponding to the ratio of the number of transconjugants/mL obtained onto selective plates to the number of viable recipient cells/mL plated in the assay, was overall comparable between the wild type and the Δ*recA* mutant for both pFLP2 and mini-CTX1. In contrast, the frequency of transconjugants that acquired and integrated the pDM4 plasmid was below the detection limit in the Δ*recA* mutant (<10^−9^), while it was basically comparable to that obtained for mini-CTX1 in the parental strain (Figure 1a). The inability of Δ*recA* to acquire the only plasmid that requires homologous recombination for chromosome integration confirmed that RecA is essential for this process in *P. aeruginosa*. To corroborate this result, we complemented the Δ*recA* mutation by integrating a copy of the *recA* gene, including its own promoter, into a neutral site of the chromosome. The resulting strain, named Δ*recA recA*^+^, showed a frequency of pDM4Δ*rsmA*-carrying transconjugants comparable to that of the wild type strain (Figure 1a).

To validate the role of RecA in the activation of the SOS response in *P. aeruginosa*, the expression of four genes, which were previously demonstrated to belong to the *P. aeruginosa* SOS regulon by independent studies [23,25,31,39,40], was evaluated in PAO1 and Δ*recA* by quantitative reverse transcription PCR (qRT-PCR). In order to set up the right conditions to induce the SOS stress response under sub-lethal doses of antibiotic, the susceptibility of the wild type and Δ*recA* strains to ciprofloxacin was preliminarily assessed by MIC assays. In line with previous reports [23,26], the Δ*recA* mutant showed a fourfold increase in ciprofloxacin susceptibility (Table 1). Bacteria were cultured in the absence and in the presence of 0.25 × MIC of ciprofloxacin and qRT-PCR analysis showed that ciprofloxacin treatment caused a 6- to 14-fold increase in the expression of SOS genes in PAO1. In contrast, no relevant increase was observed in SOS gene expression in the Δ*recA* mutant treated with ciprofloxacin (Figure 1b). This result confirmed that RecA is required for the activation of the SOS response under antibiotic-induced stress.

### 2.2. RecA Affects Intrinsic Resistance Only towards Fluoroquinolones and Genotoxic Agents

Previous studies and our preliminary analysis showed that *P. aeruginosa recA* mutants are more sensitive to ciprofloxacin than the parental strains [23,26] (Table 1). To investigate whether *recA* deletion could affect *P. aeruginosa* resistance to antibiotics other than ciprofloxacin, MIC assays were performed with *P. aeruginosa* PAO1, the Δ*recA* mutant, and the complemented Δ*recA recA*^+^ strain exposed to antibiotics belonging to different classes (Table 1). The resistance profile of Δ*recA* was overall comparable to that of the parental strain, except for the two fluoroquinolones, ciprofloxacin and ofloxacin, to which the Δ*recA* mutant showed a fourfold decrease in resistance. A two fold MIC reduction was also observed in the Δ*recA* mutant for nalidixic acid (Table 1), a quinolone-like antibiotic that is poorly active against *P. aeruginosa* [41]. Fluoroquinolones are a class of bactericidal antibiotics that target DNA gyrase and topoisomerase IV and cause DNA damage [42]. To evaluate whether the increased sensitivity of RecA deficient cells to fluoroquinolones could be attributed to the genotoxic stress induced by this class of antibiotics that, in the Δ*recA* mutant, cannot be counteracted by the RecA-mediated recombination dependent DNA repair [43], a MIC assay has been performed also with the DNA-damaging cytotoxic agent mitomycin C that has been reported to be active also against *P. aeruginosa* [44]. A fourfold reduction in the MIC value for the Δ*recA* mutant with respect to the wild type was observed (Table 1), confirming the important role of RecA in the response to genotoxic stress in *P. aeruginosa*. As expected, fluoroquinolones and mitomycin C resistance was restored to wild type levels in the complemented strain Δ*recA recA*^+^ (Table 1).

Overall, this analysis revealed that, among the antibiotics tested in this study, RecA deficiency in *P. aeruginosa* specifically leads to increased sensitivity to fluoroquinolones. However, until now the importance of RecA and/or the SOS response in the resistance to fluoroquinolones in *P. aeruginosa* has been only reported for intrinsically susceptible strains [23,26] (Table 1). We therefore wondered if this increase in sensitivity could be also observed in fluoroquinolone-resistant *P. aeruginosa* isolates. If so, the inhibition of RecA could represent a therapeutic strategy not only for potentiating the efficacy of fluoroquinolones against susceptible strains, but also for resensitizing resistant ones. To address this issue, spontaneous ciprofloxacin-resistant (CIP^R^) PAO1 mutants were selected on ciprofloxacin-containing plates and their resistance was verified by MIC assay. Two clones were selected that showed an increase in resistance corresponding to 16- or 64-fold the wild type MIC (PAO1 CIP^R^-1 and PAO1 CIP^R^-2, respectively, in Table 1). The *recA* gene was then deleted from these resistant clones, and the resulting Δ*recA* derivatives were tested for ciprofloxacin resistance by MIC assay. Even though ciprofloxacin resistance was lower in the Δ*recA* derivatives compared to their parental strains (2- or 4-fold), in line with what was previously observed for the wild type strain PAO1 (Table 1), MIC values for both strains were still above the clinical breakpoint (0.5 μg/mL) [45], implying that they have to be considered resistant to ciprofloxacin from a clinical point of view. This result suggests that RecA inhibition could not be a valuable strategy to restore sensitivity in ciprofloxacin-resistant *P. aeruginosa* isolates. However, it must be emphasized that only two resistant clones have been analyzed in this study (both deriving from the same *P. aeruginosa* strain), and that the mutations conferring resistance have not been investigated. So, we cannot rule out that RecA inhibitors could synergize with fluoroquinolones in some clinical isolates and/or in the presence of specific mutations conferring fluoroquinolones resistance.

### 2.3. RecA Is Not Required for Antibiotic-Induced Mutagenesis and Acquisition of Antibiotic Resistance

MIC assays have suggested that, except for fluoroquinolones, RecA does not influence the intrinsic level of resistance to antibiotics in *P. aeruginosa* (Table 1). However, RecA is responsible for the activation of the SOS response [23,26,46] that, in different bacteria, increases the mutation rate under stressful conditions (e.g., antibiotic treatment), mainly by inducing error-prone DNA polymerases and/or DNA repair pathways [3,5,9,47]. Therefore, the contribution of RecA to the acquisition of antibiotic resistance has also been evaluated.

First, we determined the frequency of spontaneous resistant mutants by plating PAO1 and the Δ*recA* mutant, precultured in a rich medium, onto agar plates containing inhibitory concentrations (ranging from 4× to 20 × MIC, depending on the antibiotic; see Materials and Methods for details) of four different bactericidal antibiotics, i.e., ofloxacin, gentamicin, colistin, and fosfomycin. Notably, no significant reduction in the frequency of resistant colonies to any of the tested antibiotics was observed in the Δ*recA* mutant compared to the wild type (Figure 2), confirming previous evidence that RecA does not affect the mutation rate in *P. aeruginosa* during planktonic growth under non-stressful conditions [29,32].

Given the central role of RecA in the SOS response, the next step was to evaluate whether some difference in the frequency of resistant mutants could be observed when the activation of the SOS response is induced. To better design our experimental setup, we first compared the effect of sub-MIC antibiotic concentrations on the growth rates of *P. aeruginosa* PAO1 and the Δ*recA* mutant. Both strains were cultured in MH in the absence or in the presence of 0.125 × MIC or 0.25 × MIC of gentamicin, colistin, ciprofloxacin, or ofloxacin. These antibiotics target different physiological functions, i.e., protein synthesis, cell envelope integrity and DNA replication, but share bactericidal activity towards *P. aeruginosa* [48,49]. The growth curves of PAO1 and the Δ*recA* mutant in the presence of equivalent concentrations of each antibiotic tested were overall comparable (Appendix A), suggesting that sub-MIC antibiotics impose similar stress to both strains. We then measured the effect of sub-MIC antibiotic treatment on the frequency of antibiotic-resistant mutants. To this aim, PAO1 and the Δ*recA* mutant were precultured in the presence of 0.25 × MIC of gentamicin, colistin, ciprofloxacin, or ofloxacin and then plated on agar plates supplemented with equivalent inhibitory concentrations of gentamicin, colistin, ofloxacin, or fosfomycin. Four different antibiotics, belonging to different classes, were used for resistant mutant selection in order to minimize any potential effect of antibiotic-specific resistance mechanisms. Moreover, samples precultured with a given antibiotic were not tested for mutants resistant to the same antibiotic (or to an antibiotic belonging to the same class), to avoid the possible positive selection of resistant mutants of interest during preculturing in the presence of sub-MIC antibiotics. We observed a significant increase in the frequency of resistant mutants for bacteria precultured in the presence of fluoroquinolones (ofloxacin and ciprofloxacin), for both PAO1 and the Δ*recA* mutant (Figure 2). In contrast, the frequency of mutants obtained for cultures treated with gentamicin or colistin was comparable to that of untreated cultures (Figure 2). This indicates that stress-induced mutagenesis is not activated by all bactericidal antibiotics in *P. aeruginosa*. Notably, this result is apparently in contrast with a previous report that several antibiotics, including gentamicin, can increase antibiotic resistance frequency in *P. aeruginosa* PAO1 [30]. However, in that study only resistance to rifampicin was assessed and the increases in mutant frequency were modest for most of the antibiotics tested (e.g., 2-3-fold increase for gentamicin) [30]. Such an increase in mutant frequency is much lower than that observed here for fluoroquinolone-treated cultures (ranging from 10- to 100-fold, depending on the antibiotic used for resistant mutant selection), and could be hardly appreciated in our experimental setting, where biological replicates showed a relevant variability in mutant frequency (Figure 2).

Interestingly, the increase in mutant frequency under fluoroquinolone stress with respect to the untreated controls was overall comparable between PAO1 and the Δ*recA* mutant (Figure 2). This result implies the existence of a RecA- and SOS response-independent mechanism that allows *P. aeruginosa* cells to acquire fluoroquinolone-induced antibiotic resistance. This is in contrast to what was previously reported for other bacteria, such as the model organism *E. coli* or *Vibrio cholerae*, in which RecA or SOS response inactivation abolishes the mutagenesis induced by antibiotic treatment [13,14,50]. A couple of studies have previously used *P. aeruginosa recA* mutants as a control in resistance acquisition experiments. Valencia *et al.* investigated the ability of amikacin to reduce the mutagenesis induced by ciprofloxacin in *P. aeruginosa*, using a disc diffusion assay, and reported that a *recA* mutant did not show the ciprofloxacin-resistant colonies observed at the edge of the inhibition zone for the parental strain [26]. However, we were not able to reproduce such a phenotype in our laboratory (data not shown). Another study, focused on the mutagenic effect of the antibiotic and antiprotozoal drug metronidazole in *P. aeruginosa*, reported that RecA-deficient cells are less likely to acquire amikacin and ciprofloxacin resistance upon metronidazole treatment than wild type cells [46]. However, the resistance frequencies obtained in that study were also very high for untreated cultures (up to 10^−2^ for amikacin) [46], suggesting that the result could be related to other resistance/tolerance mechanisms rather than to randomly acquired genetic mutations. It is worth noting that our results are instead in agreement with the observation that the SOS response is dispensable for *P. aeruginosa* evolvability towards high levels of ciprofloxacin resistance during long-term *in vitro* evolution experiments [27].

### 2.4. Specialized and Error-Prone DNA Polymerases Are not Involved in Fluoroquinolone-Induced Antibiotic Resistance

The mutagenesis induced by the activation of the SOS response in *E. coli* is largely due to the induction of specialized or low-fidelity DNA polymerases, which can increase the mutation rate through error-prone replication of damaged DNA [3,11,12]. *P. aeruginosa* has three specialized DNA polymerases, namely PolB (or polymerase II), DinB (or polymerase IV), and ImuBC, the last two being error-prone [10]. While the expression of ImuBC is clearly under the control of the SOS response [23,25,30] (Figure 1), the regulatory pathways responsible for the induction of the *P. aeruginosa polB* and *dinB* genes are still poorly defined. Transcriptomic analysis demonstrated that *polB* expression is not affected by ciprofloxacin, while *dinB* appeared to be induced by ciprofloxacin independently of the SOS response [23]. Accordingly, DinB-associated mutagenesis was found to be independent of RecA in the close relative *Pseudomonas putida*, although *dinB* transcription was not found to be induced by DNA damaging agents in this bacterium [51]. In contrast, Sanders and colleagues proposed that LexA can regulate *dinB* gene expression in *P. aeruginosa* and that it is able to bind to the *dinB* promoter *in vitro*, although with low affinity [52]. However, the DNA fragment used to investigate LexA binding in the electrophoretic mobility shift assay (EMSA) also contained the promoter (and LexA box) of the divergently oriented gene PA0922 [52] which belongs to the LexA regulon [23].

In order to verify whether these specialized DNA polymerases could be involved in fluoroquinolone-induced mutagenesis in *P. aeruginosa*, we first analyzed by qRT-PCR the effect of sub-MIC ciprofloxacin treatment on the expression of polymerase genes in our experimental conditions. The genes *polA* and *polC*, encoding the catalytic subunits of the replicative DNA polymerases I and III, respectively, were included in the analysis as controls. Besides the operon that encodes ImuBC (Figure 1), we observed ciprofloxacin-mediated induction of *dinB* and *polA*, while *polB* and *polC* expression was not affected by sub-MIC ciprofloxacin (Figure 3a). Notably, both *dinB* and *polA* were induced by ciprofloxacin also in the Δ*recA* mutant, implying that the antibiotic triggers the expression of these genes through an SOS-independent mechanism. While the results for *dinB* and *polB* agree with previous transcriptomic data, *polA* up-regulation in response to ciprofloxacin was not previously described [23]. However, in the transcriptomic study, cells were treated for 2 h with an inhibitory ciprofloxacin concentration (8 × MIC) [23], while here bacteria have been cultured in the presence of a subinhibitory antibiotic concentration (0.25 × MIC). The different experimental settings could therefore explain the different outcomes.

Nevertheless, our original aim was to verify whether specialized DNA polymerases are involved in the ciprofloxacin-mediated increase of the mutagenesis rate, so we investigated the effect of *polB, dinB*, or *imuC* deletion in both RecA-proficient (PAO1) and -deficient (Δ*recA*) backgrounds (Appendix A). Preliminary experiments showed that ciprofloxacin MICs were identical between polymerase-deficient strains and the corresponding control (PAO1 or Δ*recA*) (data not shown). Since the increase in resistant mutant frequency induced by fluoroquinolones was observed under all conditions tested (Figure 2), we only used gentamicin-containing plates to compare the effect of sub-inhibitory (0.25 × MIC) ciprofloxacin treatment on mutation frequency in these mutants. Notably, we observed a comparable induction of resistance frequency by ciprofloxacin in all mutants, irrespective of the presence or absence of RecA (Figure 2b). Thus, even if DinB and ImuBC were previously reported to play a role in the tolerance of *P. aeruginosa* to DNA damages, such as those induced by alkylating agents and ROS [52,53,54], and ImuBC was also found to contribute to UV-induced mutagenesis [40], our results strongly suggest that none of the specialized DNA polymerases of *P. aeruginosa* is required for the fluoroquinolone-mediated increase in the mutation rate.

We also tried to generate a *polA* deletion mutant in PAO1, but all our attempts failed, even if this gene is not deemed as essential in *P. aeruginosa* according to transposon mutagenesis studies [55,56,57]. Further experiments are therefore required to verify the relevance of PolA for *P. aeruginosa* physiology and its possible role in antibiotic-induced mutagenesis, also considering that a transposon insertion *polA* mutant of *P. aeruginosa* PAO1 was previously reported to be slightly impaired in UV-induced mutagenesis (about a threefold reduction with respect to the parental strain) [52].

## 3. Conclusions

Inhibition of the SOS response regulatory circuit has been proposed as a possible therapeutic strategy to suppress stress-induced mutagenesis and hamper the emergence of antibiotic resistance [15,16]. Due to its central role in the activation of the SOS response in bacteria, RecA has been extensively exploited in the last decade as a target for the development of SOS inhibitors [58,59,60,61]. However, while *recA* gene inactivation clearly reduces stress- and antibiotic-induced resistance in some bacteria, such *E. coli* and *V. cholerae* [13,14,50], the contribution of RecA and the SOS response to antibiotic resistance in *P. aeruginosa* was less clear. Mechanistically, *P. aeruginosa* RecA could favor the emergence of mutations and, thus, of antibiotic-resistant mutants through either its recombinational activity, by promoting mutagenic DSB repair [29], or by inducing the SOS response and error-prone polymerases, such as ImuBC and/or DinB [40,52,53,54].

Here, we demonstrated that *recA* deletion in the reference strain *P. aeruginosa* PAO1 increases sensitivity only towards fluoroquinolones and genotoxic agents, while the resistance profile to other antibiotic classes (i.e*.,* aminoglycosides, β-lactams, and polymyxins) was not affected by RecA deficiency. Moreover, antibiotic-induced mutagenesis in *P. aeruginosa* appeared to be specific to fluoroquinolones (i.e., ciprofloxacin and ofloxacin), although we did not test all antibiotic classes or other fluoroquinolones, so we cannot rule out that some other antibiotics might also display mutagenic effects. More importantly, we found that both spontaneous and fluoroquinolone-induced mutagenesis occurs at comparable rates in RecA-proficient and -deficient *P. aeruginosa* cells, and is not affected by the genetic inactivation of the error-prone polymerases DinB and ImuBC. It is known that fluoroquinolones can induce DNA damages and mutations as a direct consequence of their mode of action, i.e., inhibition of DNA gyrases and type IV-topoisomerases [3,42,62], and so we can speculate that this mechanism accounts for most of the fluoroquinolone-induced mutagenesis observed in *P. aeruginosa*. However, other experiments are required to verify this hypothesis. Irrespective of these mechanistic aspects, our results strongly suggest that the inhibition of RecA and/or of the SOS response does not represent a promising approach to reduce the emergence of mutations conferring antibiotic resistance in *P. aeruginosa*. Since our study was performed on a reference laboratory strain, the possibility that the SOS response might have a higher impact on antibiotic-induced mutagenesis in some clinical isolates cannot be excluded. On the other hand, this work corroborates previous evidence that RecA inhibitors could be useful as fluoroquinolone adjuvants against *P. aeruginosa* also. 

## 4. Materials and Methods

### 4.1. Bacterial Strains and Growth Media

Bacterial strains used in this study are listed in Appendix A. Bacteria were cultured in Lysogeny Broth, Lennox formulation (LB) for genetic manipulation, while growth assays were performed in Mueller-Hinton broth (MH). For genetic manipulation procedures, antibiotics were added at the following concentrations for *E. coli*, with the concentrations used for *P. aeruginosa* shown in brackets: ampicillin, 100 µg/mL; carbenicillin (400 µg/mL); chloramphenicol 30 µg/mL (350 µg/mL); nalidixic acid, 10–20 µg/mL; tetracycline 12 µg/mL (100 µg/mL). For mutant selection assays, antibiotics were added at the concentrations indicated in the text.

### 4.2. Generation of the Complementing Construct Mini-CTX1recA and recA Deletion Mutants

Recombinant DNA procedures have been described elsewhere [63]. The *recA* gene, including its own promoter, was PCR amplified by using the Q5 Hot Start High-Fidelity DNA Polymerase (New England Biolabs, Ipswich, MA, USA), and the genomic DNA of the *P. aeruginosa* strain PAO1 as the template. Primers and restriction enzymes used for cloning are described in Appendix A. The amplicon was directionally cloned into pBluescript II KS+ (Appendix A), sequenced and subcloned into the integration-proficient plasmid mini-CTX1 [36]. The construct mini-CTX1*recA* was verified by restriction analysis and used to introduce the *recA* gene into the neutral chromosomal site *attB* of the *P. aeruginosa* Δ*recA* mutant, as previously described [64], yielding the Δ*recA recA^+^* strain (Appendix A).

Deletion of *recA* in CIP^R^ spontaneous mutants was obtained using the pDM4 Δ*recA* derivative (Appendix A) as described [65]. Single and double deletion mutants in *recA*, *polB*, *dinB* and/or *imuC* of the reference strain PAO1 (Appendix A) were previously generated [65,66].

### 4.3. Growth and Conjugation Assays

For growth assays, bacterial strains were precultured in MH at 37 °C and then refreshed 1:1000 in the same medium in the absence or in the presence of gentamicin, colistin, ofloxacin, or ciprofloxacin at the concentration corresponding to 0.25× or 0.125 × MIC. Growth assays were performed in 96-well microtiter plates (200 μL in each well) at 37 °C in a Spark 10M microtiter plate reader (Tecan), and growth was measured over time as the optical density at 600 nm wavelength (OD_600_) of the bacterial cultures.

Conjugations were performed as previously described [66]. The frequency of transconjugants for each donor/recipient pair was calculated as the ratio between the number of transconjugant colonies obtained on selective agar plates and the total number of colony forming units (CFU) for the *P. aeruginosa* recipient strain, determined by plating serial dilutions of the conjugation mixture onto LB agar plates supplemented with 10 µg/mL nalidixic acid to counter-select the *E. coli* S17.1 λ*pir* donor strain [67].

### 4.4. Gene Expression Analysis by qRT-PCR

The expression levels of selected genes (*recN*, *recX*, *lexA*, *imuB*, *polA*, *polB*, *polC*, and *dinB*) were determined by qRT-PCR. For RNA extraction, bacterial cells were cultured in MH in the absence or in the presence of 0.25 × MIC ciprofloxacin until the mid-exponential phase, harvested by centrifugation and treated with RNA-protect Bacteria Reagent (Qiagen, Hilden, Germany). Total RNA was purified using the RNeasy Mini kit (Qiagen), treated with Turbo DNase (Ambion, Austin, TX, USA), and re-purified with the RNeasy MinElute Cleanup kit (Qiagen). cDNA was reverse transcribed from 0.5 μg of total RNA with Prime Script RT Reagent Kit (Takara, Kusatsu, Japan). The cDNA was used as the template for qRT-PCR in a AriaMx Real-Time PCR System (Agilent, Santa Clara, CA, USA) using TB Green Premier EX Taq master mix (Takara). The primers used for qRT-PCR are listed in Appendix A. Relative expression of each gene with respect to the housekeeping gene *rpoD*, encoding for the vegetative sigma factor of the RNA polymerase, was calculated using the 2^−ΔΔC*t*^ method [68].

### 4.5. MIC Assays

MIC was determined using the standard broth microdilution method. Strains were cultured in MH and refreshed at ca. 5 × 10^5^ cells/mL in MH containing increasing concentrations of mitomycin C, colistin, meropenem, ofloxacin, gentamicin, ciprofloxacin, tobramycin, or nalidixic acid. MIC values were visually assessed after 24 h of growth at 37 °C under static conditions. Each strain was tested in at least three independent experiments.

### 4.6. Selection and Frequency of Antibiotic-Resistant Mutants

To evaluate the frequency of antibiotic-resistant mutants, strains were cultured in MH broth at 37 °C until the late exponential/early stationary phase in the absence or in the presence of 0.25 × MIC of selected antibiotics. Bacterial cells were harvested by centrifugation and resuspended in sterile saline solution at OD_600_ = 1. Serial dilutions were performed and plated onto MH agar plates to measure the total number of CFU. Antibiotic-resistant mutants were selected by plating 100-μL aliquots of the undiluted samples onto MH agar plates supplemented with gentamicin at 20 × MIC, colistin at 5 × MIC, fosfomycin at 4 × MIC or ofloxacin at 5 × MIC for the specific strain to be tested (either PAO1 or Δ*recA*). Gentamicin, ofloxacin, and colistin concentrations were chosen, based on preliminary experiments, in order to obtain a frequency of resistant mutants around 10^−7^ for the wild type strain (data not shown), while fosfomycin concentration was mimicked from a previous study that investigated the emergence of fosfomycin resistance in *P. aeruginosa* [31]. Colonies obtained in the presence of antibiotics were counted after 48 h of incubation at 37 °C. The frequency of spontaneous or induced resistant mutants was calculated as the ratio between the number of antibiotic-resistant CFU (selected on antibiotic-supplemented agar plates) and the number of total CFU.

To obtain the CIP^R^ mutants, 100-μL aliquots of a bacterial suspension at OD_600_ = 1 of PAO1 wild type were plated onto MH agar plates supplemented with 1 µg/mL ciprofloxacin (corresponding to 8 × MIC). The acquisition of resistance in selected clones (*n* = 12) obtained on ciprofloxacin-containing plates was verified through MIC assays. Two clones showing a variable level of ciprofloxacin resistance (Table 1) were selected to generate the corresponding Δ*recA* derivatives (see Section 4.2) (Appendix A).

### 4.7. Statistical Analysis

A statistical analysis was performed with the software GraphPad Instat using the unpaired *t*-test.

## Figures and Tables

**Figure 1 antibiotics-11-00325-f001:**
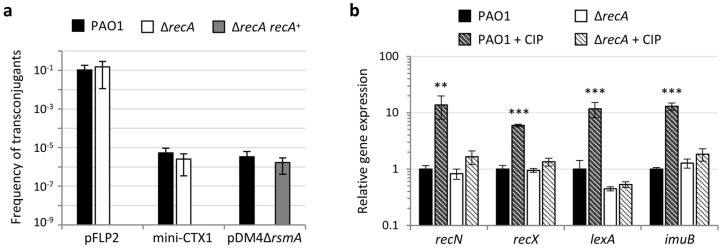
Confirmation of the essential role of RecA in homologous recombination and activation of the SOS response in *P. aeruginosa*. (**a**) Frequency of transconjugants obtained for *P. aeruginosa* PAO1, the Δ*recA* mutant, and the complemented strain Δ*recA recA*^+^ upon conjugation-mediated acquisition of the replicative plasmid pFLP2, the self-integrative suicide plasmid mini-CTX1 or the suicide plasmid pDM4Δ*rsmA*, that requires homologous recombination for chromosome integration. Frequencies are expressed as transconjugant CFUs/recipient CFUs ratio. (**b**) Relative mRNA levels of selected SOS response genes (*recN*, *recX*, *lexA*, and *imuB*) determined by qRT-PCR in *P. aeruginosa* PAO1 and the Δ*recA* mutant cultured in MH in the absence or in the presence of 0.25 × MIC ciprofloxacin (CIP) (0.031 and 0.008 μg/mL for PAO1 and Δ*recA*, respectively; Table 1). Values are the mean (± SD) of three independent experiments. Asterisks indicate a statistically significant increase in relative gene expression with respect to the untreated control (unpaired *t* test: ** *p* < 0.01; *** *p* < 0.001).

**Figure 2 antibiotics-11-00325-f002:**
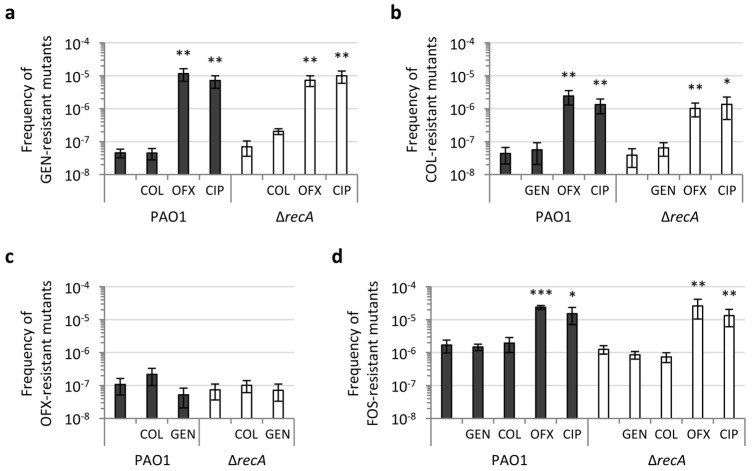
Fluoroquinolones induce mutagenesis in a RecA-independent manner in *P. aeruginosa*. Frequency of antibiotic-resistant mutants obtained on selective agar plates containing (**a**) gentamicin (GEN) at 20 × MIC, (**b**) colistin (COL) at 5 × MIC, (**c**) ofloxacin (OFX) at 5 × MIC, or (**d**) fosfomycin (FOS) at 4 × MIC, for *P. aeruginosa* PAO1 and the Δ*recA* mutant. Strains were precultured in MH at 37 °C in the absence or in the presence of GEN, COL, OFX or ciprofloxacin (CIP) at 0.25 × MIC (as indicated below each graph). Values are the mean (± SD) of at least five independent experiments. Asterisks indicate a statistically significant difference in the frequency of mutants with respect to the corresponding untreated control (unpaired *t* test: * *p* < 0.05; ** *p* < 0.01; *** *p* < 0.001). No statistically significant differences were observed between PAO1 and Δ*recA*.

**Figure 3 antibiotics-11-00325-f003:**
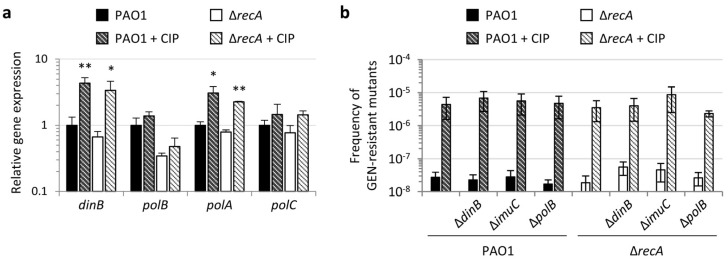
Specialized DNA polymerases are not required for fluoroquinolone-induced mutagenesis in *P. aeruginosa*. (**a**) Relative mRNA levels of the DNA polymerase genes *dinB*, *polB*, *polA*, and *polC*, determined by qRT-PCR, in *P. aeruginosa* PAO1 and the Δ*recA* mutant cultured in MH in the absence or in the presence of 0.25 × MIC ciprofloxacin (CIP) (0.031 and 0.008 μg/mL for PAO1 and Δ*recA*, respectively; Table 1). Values are the mean (± SD) of three independent experiments. Asterisks indicate a statistically significant difference in the frequency of mutants with respect to the corresponding untreated control (unpaired *t* test: * *p* < 0.05; ** *p* < 0.01). (**b**) Frequency of gentamicin (GEN) resistant mutants obtained on agar plates containing GEN at 20 × MIC for *P. aeruginosa* PAO1, the Δ*recA* mutant, and the cognate *polB*, *dinB*, and *imuC* mutants. Strains were precultured in MH at 37 °C in the absence or in the presence of CIP at 0.25 × MIC. Values are the mean (± SD) of four independent experiments. No statistically significant differences between strains were observed under the same experimental condition (absence or presence of CIP in the preinoculum).

**Table 1 antibiotics-11-00325-t001:** MIC of different compounds for the indicated *P. aeruginosa* strains.

Strain	MIC (µg/mL)							
	CIP ^1^	OFX ^1^	NAL ^1^	GEN ^1^	TOB ^1^	MER ^1^	COL ^1^	MMC ^1^
PAO1	0.125	1	250	0.5	0.25	0.5	1	4
Δ*recA*	0.031	0.25	125	0.5	0.25	0.5	1	1
Δ*recA recA*^+^	0.125	1	250	0.5	0.25	0.5	1	4
PAO1 CIP^R^-1	2	n.t. ^2^	n.t.	n.t.	n.t.	n.t.	n.t.	n.t.
PAO1 CIP^R^-1 Δ*recA*	1	n.t.	n.t.	n.t.	n.t.	n.t.	n.t.	n.t.
PAO1 CIP^R^-2	8	n.t.	n.t.	n.t.	n.t.	n.t.	n.t.	n.t.
PAO1 CIP^R^-2 Δ*recA*	2	n.t.	n.t.	n.t.	n.t.	n.t.	n.t.	n.t.

^1^ Abbreviations: CIP, ciprofloxacin; OFX, ofloxacin; NAL, nalidixic acid; GEN, gentamicin; TOB, tobramycin; MER, meropenem; COL, colistin; MMC, mitomycin C. ^2^ n.t., not tested.

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
