# Peer review of "RecA and Specialized Error-Prone DNA Polymerases Are Not Required for Mutagenesis and Antibiotic Resistance Induced by Fluoroquinolones in Pseudomonas aeruginosa"

_antibiotics, 2022, doi:10.3390/antibiotics11030325_

Round 1

Reviewer 1 Report

This is a very nicely written research article on the molecular factors involved in fluoroquinolones (FQ) resistance in Pseudomonas aeruginosa. The authors have demonstrated that the SOS response is not required for FQ resistance and that RecA is not a suitable target to develop novel therapeutic strategies against antibiotic-resistant P. aeruginosa strains. I have very no major concerns with the manuscript, the results are clear, although the figures look blurry in the submitted pdf. The only improvement that the Authors should consider is to propose some alternatives in the discussion to RecA inhibitors. If this strategy is not promising, then what is the alternative (if there is any currently)?

Author Response

Reviewer 1

Authors’ reply. First of all, we wish to thank the Reviewer for his/her positive feedback about our manuscript. Regarding the low quality of the figures, this is likely due to the fact that the figures were embedded in the word file used to generate the pdf (as requested by Antibiotics for the first submission). In the resubmission, we will upload high-resolution TIFF files and this should solve the problem. Concerning the last point, we are sorry to delude the Reviewer but, unfortunately, we cannot
propose alternatives to RecA inhibitors to hamper or delay the emergence of antibiotic-induced mutations leading to resistance. Indeed, the original aim of our work was to validate RecA as a suitable target, rather than demonstrate that it isn’t. Further work is needed to clarify how fluoroquinolones induce mutagenesis in P. aeruginosa and to verify if the underlying process/mechanism is druggable.

Reviewer 2 Report

Abstract :

The abstract is well written and summarizes at a glance the aims, results, and conclusions of this work. The methods are not well outlined in the abstract, despite the fact that they may be derived or inferred from the results. I would recommend broadly reflecting on the nature of experimental work in the abstract itself (e.g., conjugations assays, cloning, gene expression analysis).  This would serve to adequately summarize your work in content and scope.     

Introduction:

  • Please rephrase the sentence from lines 39-42. The sentence starts with “In many bacteria…. in response to DNA damage”. Although I am able to gather what is implied, it is confusing to read due to a grammatical issue.
  • The introduction sufficiently highlights the SOS gene derepression pathway via LexA-mediated autolysis as a response to DNA damage; setting the stage for the potential role of drugs targeting RecA (with an aim to reduce emergence of resistance strains).
  • Would recommend rephrasing lines 63-64, specifically in reference to the usage of the term “[intrinsic] resistance”. Gene mutations conferring gain or loss of function impacting outer membrane porin and/or efflux pumps are often necessary to achieve clinically significant antibiotic resistance, it would be inaccurate to characterize these mechanisms as ‘innate’. For instance, Pseudomonas strains with wild type expression levels of OprD or MexAbOprM are susceptible to commonly tested cephalosporins.  
  • Grammar syntax lines 65-75. Please proofread and correct the multiple tense and other grammatical errors within this section.

Results and discussion

  • In the first section, the role of RecA in homologous recombination and functional activation of the SOS response in P. aeruginosa is clearly shown. The box and whisker plot shown in figure 1 appears as low image quality. Please attempt to reproduce higher DPI (desired 600 dpi) or resolution. Perhaps use the original image for the upload and avoid screenshots/copy-paste function to avoid blurry output. It is difficult to read the exponents power of 10 on the Y axis especially in figure 1a.
  • Regarding mRNA levels of selected genes recN, recX, lexA, and imuB
    • Rationale for choice?
    • Technical comment: The asterisk markup indicating significant increases in gene expression are blurry and unclear. This pertains to the preceding image quality issue
  • Please discuss further the choice of plasmid in the context of anticipated presence or absence of other SOS gene regulators which may interfere with these results. For instance, umuD was identified and located on the conjugative plasmid pUM505. UmuD participates in the regulation of SOS gene expression. Notably, umuDC-like sequences are reported more than anticipated in This is a relevant potential confounder of results when evaluating SOS gene expression.

  • Starting with line 140 and through the ensuing discussion into section 2.2, would recommend considering not using the expression “increased susceptibility” because it commonly generates confusion. When reading through text and side glancing onto tables, one read “increased” susceptibility but has to be mentally cognizant that this implies decreased MICs. A four-fold increase in susceptibility reflected by a 4X decrease in MIC from 0.125 to 0.031 may at-a-glance confuse readers. The opposite is not true, since “increased resistance” reflects increased MICs which can be followed effortlessly.  Although this is not an error on your part, and you may certainly decide to leave this unchanged. “Lower or decreased MICs” is a potential replacement term that is understood unanimously by scientists and clinicians.  In contrast, around line 159, you used the term “2-fold’ MIC reduction when discussing nalidixic acid, which is definitely easier to follow.

  • Please comment on the mechanistic choice of quinolones Ciprofloxacin and Ofloxacin. The assays performed in section 2.2 could have been employed to evaluate levofloxacin, which is a clinically relevant antipseudomonal fluoroquinolone. In fact, the clinical breakpoint for levofloxacin is higher than that of ciprofloxacin making it a good candidate to develop PAO1 LevR-1 and -2. As antipseudomonal fluoroquinolone agents, an MIC of 1 μg/mL is considered resistant for ciprofloxacin but susceptible for levofloxacin.

  • In general, would recommend adding a word of caution when introducing results and discussing potential novel therapies within the context of induced quinolone susceptibility via Rec-mediated/genotoxic stress. Generally, would iterate that this testing involves lab-adapted reference strains as opposed to clinical isolates. This similarly applies to the discussion of therapeutic alternatives. This does not take anything from the novelty of your results and methodology; but you highlight to readers and acknowledge this limitation.

  • Lines 184-197: interesting rationale to test your hypothesis on induced ciprofloxacin resistance in PAO1. How were PAO1 CIPR-1 and PAO1 CIPR-2, chosen? Why did they greatly differ in their initial ciprofloxacin MIC prior to any ΔrecA modulation?

  • Looking at Table 1 within the context of the discussion from 189-197; I would state outright that ΔrecA derivatives exhibited lower MICs to ciprofloxacin compared to their corresponding parental strains. It is a fact that MICs for ΔrecA derivatives remain above both CLSI and EUCAST clinical breakpoints; however, it is also a fact that you have reduced the MIC of PAO1 CIPR-2 ΔrecA 4-fold. You have also reduced the MIC of PAO1 CIPR-1 ΔrecA from 2 to 1, leaving it within a dilution of clinical susceptibility. A logical question that consequently comes to mind is: what would happen if your parent strain had an MIC of 1 μg/mL or 0.75 μg/mL? Both those starting MICs are considered resistant according to clinical breakpoints. Following the pattern seen in Table 1, I could extrapolate that RecA inhibition may decrease MICs into the susceptible ranges. For that reason, I would not be too hasty to label this strategy “not valuable”, as you have in lines 191-192. This is a major standalone finding that paves the way for future work.

  • Please comment or hypothesize on why the decrease in MIC was significantly greater for PAO1 CIPR-2 ΔrecA corresponding to its ‘parent’ strain as compared to the drop seen in PAO1 CIPR-1 ΔrecA as compared to its parent strain. This is important to highlight even if the reasons are not particularly apparent because it suggests heterogeneity in applying this strategy. After all, had the MIC of PAO1 CIPR-1 dropped by the same proportion that PAO1 CIPR-2 did, the derivative strain PAO1 CIPR-1 ΔrecA MIC would have landed in the susceptible range. Was there anything particular about PAO1 CIPR-1 itself or the methodology of clone selection that would explain the smaller drop in MIC drop (i.e., from 2 to 1 μg/mL)?

  • Lines 208-216: please comment on the choice of antimicrobial agents gentamicin, fosfomycin, colistin, and ofloxacin.

  • Line 252: Change the expression ‘fold-increase’ as it doesn’t translate into the intended meaning. Perhaps use terms like magnitude or scale.

  • The same comments for figure 1 apply to figure 2. Image needs to be significantly of higher resolution.

  • Please discuss the potential reasons for the similarity seen in mutant frequency between PAO1 and ΔrecA mutants under fluoroquinolone stress.  Please highlight and discuss other RecA-independent mutations or mechanisms that may account for SOS response activation prior to section 2.4 and discussion of PolB, DinB, and ImuBC.

  • Line 315: Please highlight and emphasize polA up-regulation in response to subinhibitory MICs of ciprofloxacin. This has not been published before within the framework of pseudomonas and deserves emphasis due to clinical implications related to dosing. The publication and assays conducted by Cirz RT et al. are not contradictory to your work since the ciprofloxacin MICs are significantly high and therefore not pertinent to evaluation of mutagenesis.
  • Agree on the analysis and progression which ultimately dissociates specialized DNA polymerases from fluoroquinolone-mediated mutation rate increase.

Concluding remarks/Conclusion

  • This study is important. Please make sure to state the novel findings clearly in the conclusion starting line 362. Your main conclusion from your work should precede the restated literature context. I would start out by mentioning that your work successfully demonstrated that recA deletion reduced MICs of Ciprofloxacin. Would then specifically list the other antibiotics that were unaffected by RecA deficiency. Would not generalize that RecA deficiency does not impact “all other classes” because this study only surveyed a few other antibiotics. Please be specific and list out the MICs of antibiotics that were unaffected. If the data to other antibiotics was not shown, please add since this is important data that must not be limited or generalized.  The third major finding is that antibiotic-induced mutagenesis in aeruginosa was shown to be specific to Ciprofloxacin. I do not think we can generalize this to all fluoroquinolones and this must be states as such in the conclusion. In the clinical setting, discordance between cipro and levo resistance is seen. The fourth major finding of this work centers around the spontaneity of mutagenesis brought about by exposure to quinolones. This is very important and carries clinical implications for therapy, infection control, and stewardship. I would therefore mention this succinctly in the conclusion section since this mutagenesis was shown to occur in RecA-proficient as well as deficient P. aeruginosa cells. The latter finding is sufficient standalone without the need to restate the autonomy of this finding from error-prone polymerases. Would focus on showing the utility of this result rather than restating other aspects that appeared in your discussion. (i.e., no need to repeat results pertaining to DinB and ImuBC in the conclusion).

Author Response

Comments and Suggestions for Authors
Abstract:
The abstract is well written and summarizes at a glance the aims, results, and conclusions of this work. The methods are not well outlined in the abstract, despite the fact that they may be derived or inferred from the results. I would recommend broadly reflecting on the nature of experimental work in the abstract itself (e.g., conjugations assays, cloning, gene expression analysis). This would serve to adequately summarize your work in content and scope.
Authors’ reply. The abstract has been modified according to the Reviewer’s suggestions and considering the 200 words limit of Antibiotics.
Introduction:
Please rephrase the sentence from lines 39-42. The sentence starts with “In many bacteria…. in response to DNA damage”. Although I am able to gather what is implied, it is confusing to read due to a grammatical issue.
Authors’ reply. The original sentence lacked the verb; we are sorry for the mistake. Now the text has been corrected (line 43 in the marked-up version of the revised manuscript).
The introduction sufficiently highlights the SOS gene derepression pathway via LexA-mediated autolysis as a response to DNA damage; setting the stage for the potential role of drugs targeting RecA (with an aim to reduce emergence of resistance strains).
Authors’ reply. We wish to thank the Reviewer for the positive comment.
Would recommend rephrasing lines 63-64, specifically in reference to the usage of the term “[intrinsic] resistance”. Gene mutations conferring gain or loss of function impacting outer membrane porin and/or efflux pumps are often necessary to achieve clinically significant antibiotic resistance, it would be inaccurate to characterize these mechanisms as ‘innate’. For instance, Pseudomonas strains with wild type expression levels of OprD or MexAbOprM are susceptible to commonly tested cephalosporins.
Authors’ reply. According to the Reviewer’s suggestion, we have deleted the word “intrinsic” (see line 66 in the marked-up version of the revised manuscript).
Grammar syntax lines 65-75. Please proofread and correct the multiple tense and other grammatical errors within this section.
Authors’ reply. The text has been revised (see lines 71-79 in the marked-up version of the revised manuscript).
Results and discussion
In the first section, the role of RecA in homologous recombination and functional activation of the SOS response in P. aeruginosa is clearly shown. The box and whisker plot shown in figure 1 appears as low image quality. Please attempt to reproduce higher DPI (desired 600 dpi) or resolution. Perhaps use the original image for the upload and avoid screenshots/copy-paste function to avoid blurry output. It is difficult to read the exponents power of 10 on the Y axis especially in figure 1a.
Authors’ reply. We are sorry for the low quality of the figures, that is likely due to the fact that the figures were embedded in the word file used to generate the pdf (as requested by Antibiotics for the first submission). In the resubmission, we will upload high-resolution TIFF files and this should solve the problem.
Regarding mRNA levels of selected genes recN, recX, lexA, and imuB. Rationale for choice?
Authors’ reply. We selected genes that were demonstrated to be regulated by the SOS response in P. aeruginosa by several independent studies. The text has been modified to make this choice clearer (see lines 138-139 in the marked-up version of the revised manuscript).
Technical comment: The asterisk markup indicating significant increases in gene expression are blurry and unclear. This pertains to the preceding image quality issue
Authors’ reply. As anticipated above, we will upload high-resolution TIFF files for the resubmission.
Please discuss further the choice of plasmid in the context of anticipated presence or absence of other SOS gene regulators which may interfere with these results. For instance, umuD was identified and located on the conjugative plasmid pUM505. UmuD participates in the regulation of SOS gene expression. Notably, umuDC-like sequences are reported more than anticipated in This is a relevant potential confounder of results when evaluating SOS gene expression.
Authors’ reply. In our work, we only used plasmids to evaluate the efficiency of the homologous recombination process (Figure 1a). The plasmids used in this assay are well known engineered plasmids (Table S1) that do not contain genes involved in the SOS response. All the other analyses, including gene expression assays, have been performed using plasmidless strains. Please note that also the complementation of the recA deletion mutant has been achieved by introducing the recA gene into the chromosome (and removal of the backbone plasmid).
Starting with line 140 and through the ensuing discussion into section 2.2, would recommend considering not using the expression “increased susceptibility” because it commonly generates confusion. When reading through text and side glancing onto tables, one read “increased” susceptibility but has to be mentally cognizant that this implies decreased MICs. A four-fold increase in susceptibility reflected by a 4X decrease in MIC from 0.125 to 0.031 may at-a-glance confuse readers. The opposite is not true, since “increased resistance” reflects increased MICs which can be followed effortlessly. Although this is not an error on your part, and you may certainly decide to leave this unchanged. “Lower or decreased MICs” is a potential replacement term that is understood unanimously by scientists and clinicians. In contrast, around line 159, you used the term “2-fold’ MIC reduction when discussing nalidixic acid, which is definitely easier to follow.
Authors’ reply. According to the Reviewer’s suggestion, we have replaced “increase in susceptibility” with “decrease in resistance” (see line 164 in the marked-up version of the revised manuscript).
Please comment on the mechanistic choice of quinolones Ciprofloxacin and Ofloxacin. The assays performed in section 2.2 could have been employed to evaluate levofloxacin, which is a clinically relevant antipseudomonal fluoroquinolone. In fact, the clinical breakpoint for levofloxacin is higher than that of ciprofloxacin making it a good candidate to develop PAO1 LevR-1 and -2. As antipseudomonal fluoroquinolone agents, an MIC of 1 μg/mL is considered resistant for ciprofloxacin but susceptible for levofloxacin.
Authors’ reply. We agree with the Reviewer that also levofloxacin could have been used. However, our aim was just to verify whether the increase in susceptibility caused by recA deletion was restricted to ciprofloxacin or common to other quinolones and/or agents that cause genotoxic stress. For this reason, we also determined the MIC for the highly active fluoroquinolone ofloxacin and the poorly active quinolone nalidixic acid, together with the genotoxic compound mitomycin C.
In general, would recommend adding a word of caution when introducing results and discussing potential novel therapies within the context of induced quinolone susceptibility via Rec-mediated/genotoxic stress. Generally, would iterate that this testing involves lab-adapted reference strains as opposed to clinical isolates. This similarly applies to the discussion of therapeutic alternatives. This does not take anything from the novelty of your results and methodology; but you highlight to readers and acknowledge this limitation.
Authors’ reply. According to the Reviewer comment, we have added a sentence at the end of the conclusion to discuss this point (see lines 395-397 in the marked-up version of the revised manuscript). Such limitation was already discussed in the Results and Discussion section (lines 197-202 in the marked-up version of the revised manuscript).
Lines 184-197: interesting rationale to test your hypothesis on induced ciprofloxacin resistance in PAO1. How were PAO1 CIPR-1 and PAO1 CIPR-2, chosen? Why did they greatly differ in their initial ciprofloxacin MIC prior to any ΔrecA modulation?
Authors’ reply. By selection on ciprofloxacin-containing plates we obtained several independent spontaneous mutants resistant to ciprofloxacin. The MIC of ciprofloxacin for twelve of these mutants was determined and the two that showed the highest and the lowest MIC among the resistant clones were selected and subjected to recA deletion mutagenesis. So, the rationale was to analyze two clones with highly variable resistance levels, just to check if the impact of recA deletion was influenced by the initial resistance level.
Looking at Table 1 within the context of the discussion from 189-197; I would state outright that ΔrecA derivatives exhibited lower MICs to ciprofloxacin compared to their corresponding parental strains. It is a fact that MICs for ΔrecA derivatives remain above both CLSI and EUCAST clinical breakpoints; however, it is also a fact that you have reduced the MIC of PAO1 CIPR-2 ΔrecA 4-fold. You have also reduced the MIC of PAO1 CIPR-1 ΔrecA from 2 to 1, leaving it within a dilution of clinical susceptibility. A logical question that consequently comes to mind is: what would happen if your parent strain had an MIC of 1 μg/mL or 0.75 μg/mL? Both those starting MICs are considered resistant according to clinical breakpoints. Following the pattern seen in Table 1, I could extrapolate that RecA inhibition may decrease MICs into the susceptible ranges. For that reason, I would not be too hasty to label this strategy “not valuable”, as you have in lines 191-192. This is a major standalone finding that paves the way for future work.
Authors’ reply. In our discussion we clearly stated that resistance was lowered by recA deletion (lines 191-193 in the marked-up version of the revised manuscript) and then discussed that the MIC values were still higher that the clinical breakpoint (line 194). On this basis, we suggested that RecA inhibition could not be a valuable strategy to resensitize resistant isolates (lines 196-197). However, in the following sentences we clearly discussed that this should be verified in clinical isolates (lines 197-202). Moreover, in line with the Reviewer’s suggestion, in the last sentence of our Conclusions we state that “RecA inhibitors could be useful as fluoroquinolone adjuvants also against P. aeruginosa.”
Please comment or hypothesize on why the decrease in MIC was significantly greater for PAO1 CIPR-2 ΔrecA corresponding to its ‘parent’ strain as compared to the drop seen in PAO1 CIPR-1 ΔrecA as compared to its parent strain. This is important to highlight even if the reasons are not particularly apparent because it suggests heterogeneity in applying this strategy. After all, had the MIC of PAO1 CIPR-1 dropped by the same proportion that PAO1 CIPR-2 did, the derivative strain PAO1 CIPR-1 ΔrecA MIC would have landed in the susceptible range. Was there anything particular about PAO1 CIPR-1 itself or the methodology of clone selection that would explain the smaller drop in MIC drop (i.e., from 2 to 1 μg/mL)?
Authors’ reply. As indicated in the manuscript, we did not characterize the ciprofloxacin-resistant mutants at the genetic level and did not investigate this issue further in our manuscript (see lines 197-202 in the marked-up version of the revised manuscript). Thus, any hypothesis to explain the slightly different effect of recA deletion in the two ciprofloxacin-resistant mutants would be too speculative at this stage, and we prefer not to include it in our manuscript.
Lines 208-216: please comment on the choice of antimicrobial agents gentamicin, fosfomycin, colistin, and ofloxacin.
Authors’ reply. We selected antibiotics that target different physiological functions (i.e. protein synthesis, cell envelope integrity and DNA replication) but share bactericidal activity towards P. aeruginosa, in order to avoid different effects potentially due to the different activity (bacteriostatic or bactericidal) of the antibiotics. This point has been explained in the manuscript (see lines 216-217 and 228-232 in the marked-up version of the revised manuscript).
Line 252: Change the expression ‘fold-increase’ as it doesn’t translate into the intended meaning. Perhaps use terms like magnitude or scale.
Authors’ reply. According to the Reviewer’s request, we deleted the word “fold” and left only the word “increase” (see line 257 in the marked-up version of the revised manuscript). We cannot use “magnitude” or “scale” as the sentence refers to the increase with respect to untreated cultures.
The same comments for figure 1 apply to figure 2. Image needs to be significantly of higher resolution.
Authors’ reply. As anticipated above, we will upload high-resolution TIFF files for the resubmission, and this is expected to solve the problem.
Please discuss the potential reasons for the similarity seen in mutant frequency between PAO1 and ΔrecA mutants under fluoroquinolone stress. Please highlight and discuss other RecA-independent mutations or mechanisms that may account for SOS response activation prior to section 2.4 and discussion of PolB, DinB, and ImuBC.
Authors’ reply. The similarity in mutant frequency between PAO1 and recA mutants is likely due to the minor contribution of the SOS response and of error-prone polymerases to mutagenesis in P. aeruginosa. This point has been discussed in lines 384-387 (lines refer to the marked-up version of the revised manuscript). A possible RecA- and DNA polymerases-independent mechanisms of ciprofloxacin-induced mutagenesis has been proposed, although it has been discussed in the Conclusions section (see lines 387-392). Indeed, we believe that it is more appropriate to provide and discuss such hypothesis after presenting (and excluding) the role of the SOS response and error-prone DNA polymerases, which are main effectors of ciprofloxacin-induced mutagenesis in the model organism E. coli.
Line 315: Please highlight and emphasize polA up-regulation in response to subinhibitory MICs of ciprofloxacin. This has not been published before within the framework of pseudomonas and deserves emphasis due to clinical implications related to dosing. The publication and assays conducted by Cirz RT et al. are not contradictory to your work since the ciprofloxacin MICs are significantly high and therefore not pertinent to evaluation of mutagenesis.
Authors’ reply. The induction of polA by ciprofloxacin has been described and discussed with respect to available literature in lines 319-325 (lines refer to the marked-up version of the revised manuscript). We agree with the Reviewer that our data are not in contrast with those by Cirz et al., as stated in lines 321-325. Moreover, the possible role of PolA in mutagenesis has been briefly discussed in lines 361-365.
Agree on the analysis and progression which ultimately dissociates specialized DNA polymerases from fluoroquinolone-mediated mutation rate increase.
Authors’ reply. We wish to thank the Reviewer for the positive comment.
Concluding remarks/Conclusion
This study is important. Please make sure to state the novel findings clearly in the conclusion starting line 362. Your main conclusion from your work should precede the restated literature context. I would start out by mentioning that your work successfully demonstrated that recA deletion reduced MICs of Ciprofloxacin. Would then specifically list the other antibiotics that were unaffected by RecA deficiency. Would not generalize that RecA deficiency does not impact “all other classes” because this study only surveyed a few other antibiotics. Please be specific and list out the MICs of antibiotics that were unaffected. If the data to other antibiotics was not shown, please add since this is important data that must not be limited or generalized. The third major finding is that antibiotic-induced mutagenesis in aeruginosa was shown to be specific to Ciprofloxacin. I do not think we can generalize this to all fluoroquinolones and this must be states as such in the conclusion. In the clinical setting, discordance between cipro and levo resistance is seen. The fourth major finding of this work centers around the spontaneity of mutagenesis brought about by exposure to quinolones. This is very important and carries clinical implications for therapy, infection control, and stewardship. I would therefore mention this succinctly in the conclusion section since this mutagenesis was shown to occur in RecA-proficient as well
as deficient P. aeruginosa cells. The latter finding is sufficient standalone without the need to restate the autonomy of this finding from error-prone polymerases. Would focus on showing the utility of this result rather than restating other aspects that appeared in your discussion. (i.e., no need to repeat results pertaining to DinB and ImuBC in the conclusion).
Authors’ reply. We are grateful to the Reviewer for the precious comments. We have modified the Conclusions according to most of his/her suggestions. However, we believe that a very brief introduction summarizing the relevant literature and the aim of the work can help the reader to follow and appreciate the main conclusions of our study.